# Application of a Machine Learning Algorithms in a Wrist-Wearable Sensor for Patient Health Monitoring during Autonomous Hospital Bed Transport

**DOI:** 10.3390/s21175711

**Published:** 2021-08-25

**Authors:** Yan Hao Tan, Yuwen Liao, Zhijie Tan, King-Ho Holden Li

**Affiliations:** School of Mechanical and Aerospace Engineering, Nanyang Technological University, Singapore 639798, Singapore; yh.tan@ntu.edu.sg (Y.H.T.); m180139@e.ntu.edu.sg (Y.L.); tanz0152@e.ntu.edu.sg (Z.T.)

**Keywords:** machine learning, polynomial regression, remote automated monitoring, wristband sensor, Dreyfus

## Abstract

Smart sensors, coupled with artificial intelligence (AI)-enabled remote automated monitoring (RAMs), can free a nurse from the task of in-person patient monitoring during the transportation process of patients between different wards in hospital settings. Automation of hospital beds using advanced robotics and sensors has been a growing trend exacerbated by the COVID crisis. In this exploratory study, a polynomial regression (PR) machine learning (ML) RAM algorithm based on a Dreyfusian descriptor for immediate wellbeing monitoring was proposed for the autonomous hospital bed transport (AHBT) application. This method was preferred over several other AI algorithm for its simplicity and quick computation. The algorithm quantified historical data using supervised photoplethysmography (PPG) data for 5 min just before the start of the autonomous journey, referred as pre-journey (PJ) dataset. During the transport process, the algorithm continued to quantify immediate measurements using non-overlapping sets of 30 PPG waveforms, referred as in-journey (IJ) dataset. In combination, this algorithm provided a binary decision condition that determined if AHBT should continue its journey to destination by checking the degree of polynomial (DoP) between PJ and IJ. Wrist PPG was used as algorithm’s monitoring parameter. PPG data was collected simultaneously from both wrists of 35 subjects, aged 21 and above in postures mimicking that in AHBT and were given full freedom of upper limb and wrist movement. It was observed that the top goodness-of-fit which indicated potentials for high data accountability had 0.2 to 0.6 cross validation score mean (CVSM) occurring at 8th to 10th DoP for PJ datasets and 0.967 to 0.994 CVSM at 9th to 10th DoP for IJ datasets. CVSM was a reliable metric to pick out the best PJ and IJ DoPs. Central tendency analysis showed that coinciding DoP distributions between PJ and IJ datasets, peaking at 8th DoP, was the precursor to high algorithm stability. Mean algorithm efficacy was 0.20 as our proposed algorithm was able to pick out all signals from a conscious subject having full freedom of movement. This efficacy was acceptable as a first ML proof of concept for AHBT. There was no observable difference between subjects’ left and right wrists.

## 1. Introduction

Developing highly skilled nurses, often called Advanced Practice Nurses (APNs), has been identified as a strategic thrust for Singapore to deliver healthcare in the community [1]. In contrast to hospitals as first source of healthcare services, the population under this healthcare paradigm would age gracefully in their respective locale of residence. First, this can be achieved through healthy living with good diet and exercise, followed by strong community care with medical checks and consultations administered by travelling medical personnel or satellite facilities. Consequently, ailments requiring hospital resources would be reduced and more medical decisions must be made beyond hospital settings. To ensure subsisting medical services are made affordable for the masses, APNs had received particular attention as alternative to doctors for their unique authority to make autonomous medical decisions in a travelling capacity. However, the number of APNs are historically small as services requiring APN had not been ubiquitously large and present training pathways require long-term commitment to a nursing career. In Singapore’s context, there remain substantial challenges to attract, train and retain APNs.

Smart functions such as remote automated monitoring (RAM) [2,3], defined as clinical healthcare measurements and assessments at a distance [3], seeks to address part of this challenge in two ways. First, relieving a nurse from on-the-job tasks (OJT) that they had attained mastery but are essential to patient outcomes. Mastery, in the Dreyfusian skill acquisition model [4], can be unique via an individual’s exceptional level of motivation and natural talent. Moreover, mastery can be observable via individual’s consistent high performances under different situations and perpetual development of similarly high performance modes. For a nurse, he or she could better assess if they had mastered an OJT and thus decide whether it is beneficial to continue training or move onto other OJT. An upward movement of nursing talent is therefore expected, generating a manpower gap for OJTs that can be mastered by most nurses. Smart functions’ second utility is to fill this gap by forming a strong in-hospital automation foundation. RAM, in particular, facilitates medical decision making from physiological data gathered from sensors monitoring a remote subject. Identified as a promising technology, there are rooms for research and development of integration with artificial intelligence (AI) and machine learning (ML), infrastructure, reliability, cyber security and data privacy [3,5]. Eventually, smart functions like RAM can free up nurse (s) from generalized rule-based routines, enabling them to focus on activities that are pathways to APN such as holistic patient care planning.

Hospital bed transport (HBT) is an OJT identified with high potential for smart function automation. HBT refers to a low-risk variant described in the body of robotic bed mover research [6,7] where a transported patient is suitable for unsupervised movement. This contrasts against acute patient bed transport that definitely requires immediate patient supervision and intervention during complications. The body of HBT research hence primarily focused on eliminating musculoskeletal work hazards for HBT personnel via powered bed movers or bed redesigns with the state of art being a semi-autonomous hospital bed [6,7,8,9]. However, present solutions still required a nurse to be in the vicinity, which meant committing time to the task which for a multi-story, multi-building hospital setting, can be up to 45 min per one-way HBT. Therefore, to substantially realize benefits from relieving a nurse from HBT, a nurse’s monitoring and diagnostic capabilities must also be automated.

The aim is to develop a RAM solution for assessing HBT patient’s immediate health monitoring. Under this solution, real-time advisories on patient’s general wellbeing will be conveyed remotely to a nurse. This is the very first time that RAM solution, together with HBT patient monitoring is introduced. Extending the full capabilities of RAM for HBT application, Dreyfusian philosophy and human skill acquisition model [4,10,11,12] were used in the studies. They provided the foundational principles through artificial intelligence (AI) approach and seminal nursing skills acquisition behavioral model [13]. In particular, a Dreyfusian descriptor from Novice to Expert ([13], p. 21) was used to prescribe ML algorithm’s three constituents: (1) historical data; (2) immediate measurement; (3) binary decision condition. In terms of RAM database, photoplethysmography (PPG) was used as a continuous wellbeing monitoring parameter. First, PPG had seen widespread acceptance in intensive care units [14,15,16,17,18] and smart health functions in consumer wearable sensors. Second, PPG-based solutions can keep cybersecurity threats to a minimum by being isolated in a finger clip or wristband form factor. Third, PPG is sufficient for patient monitoring as it detects blood flow in veins and blood oxygen content. This preserves patient privacy from medical diagnostic information with high resolution measurements such as electrocardiogram (ECG) [19].

Our proposed method uses PPG measurements to quantify cardiac-induced blood volumetric changes in a patient’s veins as a sign of his/ her wellbeing. With this critical information, the decision-making process of proceeding of AHBT’s journey to its final destination is based on the correlation of the monitored patient’s current PPG data with his or her previously supervised measurement. Four reported AI algorithms were studied for suitability in the ABHT application in this work. First was single feature polynomial regression [20], which was easy to communicate for operator acceptance and it treated all measured PPG data with equal significance at the cost of being more susceptible to noise inputs. Second, rule-based PPG signal quality assessment [21,22] which did not require any data annotation at the cost of deploying complex layers of decision-making rules that were difficult for operator acceptance. Third, unsupervised multi-feature anomaly detection [23] which could perform for different activities at cost of uncertainty due to several complex features driving a decision. Finally, an end-to-end deep learning [24] which showed exceptional quantified results at cost of inexplicability and patient relevancy by cross applying data from other subjects, application context or patients. Details of these various methods are covered in Section 2 of this paper.

The objective of this first of its kind research is an exploratory study of the proposed PPG RAM ML algorithm for HBT. The scope of this study will focus on algorithm’s efficacy to transport a patient without false positives or stoppages via goodness-of-fit, central tendencies and efficacy metrics. PPG wristband sensors were used across a statistically significant population (*N* = 35) of 21 years and above, building upon our earlier pilot study [20] which showed up to 95% algorithm efficacy using a single subject’s left middle finger PPG across statistically significant number of runs. Measurements were taken simultaneously on left and right wrists to determine past research observations of having better signal quality from non-dominant wrist [24].

Section 2 of this paper presents methods and materials detailing algorithm design followed by data collection procedures. Section 3 covers the results and analysis. Section 4 concludes the study, highlighting significance to HBT and in-community healthcare.

## 2. Algorithm Design, Experimental Setup and Data Collection

### 2.1. Dreyfusian AI Algorithm Design

Our proposed algorithm was based on a Dreyfusian descriptor of a *Novice* nurse during monitoring and diagnostic work to maximize functions identified for automation ([13], p. 10). In consideration of AHBT concept of operations, the three constituents are: (1) historical data; (2) immediate measurement (s); (3) binary decision condition which were identified and translated from descriptor to algorithm specifications, as shown in Table 1.

First, historical data of the subject was defined as known priori of his/ her normal condition. For AHBT, supervised and authorized PPG measurement was taken for 5 min just prior to the start of autonomous journey, hence pre-fixed as pre-journey (PJ). These quantifications had taken into consideration of both operational needs and statistical requirement as PJ phase should not be too short where waveforms per dataset would fall below statistical significance under central limit theorem (*N* = 30). Second, immediate measurement was defined as obtained information of patient immediate wellbeing. Specifically, they were the latest, continuous and non-overlapping sets of 30 PPG waveforms during AHBT travel towards its destination, hence pre-fixed as in-journey (IJ). Similarly, 30 PPG waveforms per IJ dataset were required for statistically significant data in the shortest possible time.

There are several AI techniques that could potential be employed in AHBT. A weighted design matrix was used to select an AI technique that is most suitable for this. These techniques were either known to be effective in modelling time-series data or had demonstrated real-time health monitoring potential on subjects, as summarized in Table 2. These included unsupervised single-feature polynomial regression (PR) [20], rule-based PPG signal quality assessment [21], unsupervised multi-feature anomaly detection [22], and end-to-end deep learning [23].

Matrix factors were selected based on AHBT operational needs. The selected AI techniques must fulfill several basic criteria. First, it must be easily understood mathematically by a nurse for field acceptance. Then, its method must be explicable for output decisions to be put to use. For example, medical AI research had recently coined the term explicable artificial intelligence (XAI) [25] to highlight the need for deep learning techniques and working principles to be understood by humans. Third, decision making algorithm must use data sets of a subject without cross referencing to other patients on the basis that care must be customized for any individual. Due to this constraint, it was necessary to ensure that the variation PPG measurement across different patients should not hinder the decision-making process. Deep learning algorithms have the disadvantage of requiring cross referencing of data to substitute large amount of data annotation while rule-based techniques do not. Ideally, there should not be data annotation throughout this decision-making process. Computation time should not be more than a couple of seconds in allowing AHBT to move continuously. Moreover, hardware which were mounted on AHBT should be a standalone system for local data processing with defense capability against cyber-attacks.

Weightage for each consideration factor were first evenly distributed at 20% each. They were further adjusted to allow heavier weightage on “easy to understand” while reducing the need for “computational time”. This was important to ensure that the nursing profession was receptive to AHBT concept. Whereas for computation time, it was foreseeable that advancement of microprocessors would bring forth improving computational capability over time and future processors would be further miniaturized.

Polynomial regression (PR) ML technique was found to be best suited for AHBT. In terms of easy to understand and explicability factors, PR was the easiest to understand due to mathematical simplicity with the concept of “line of best fit” which most nurses’ learnt in their math curriculum. Single feature in degree-of-polynomial (DoP), defined as the highest exponent power of a modelled PR function, was also easy to communicate. There were some challenges involved communication of ML specific concepts such as train-test split and cross validation. Other techniques were deemed unsuitable due to their abstract mathematical concepts, multiple features or AI deep learning blackbox [25] and hence they scored lower relative to PR.

In terms of data and computation factors, PR, rule-based and multi-feature techniques required only raw PPG measurements from the subject himself to function, while deep learning required the use of multiple public datasets in order to mitigate large amount of annotations. Annotations were still present in PR and multi-feature techniques but were limited only binary “good” or “bad” classifications. Rule based was the best option in this regard as it did not require any annotation work while deep learning inherently required extensive annotations.

Computational time were scored based on complexity of their algorithm. Each of them had their reported hardware and relative performance [21,22]. For PR [20], implementation was conducted with a laptop with intel i5-8250U processor (Intel Corporation, Santa Clara, CA, USA) and NVIDIA GeForce MX150 GPU (Nvidia Corporation, Santa Clara, CA, USA) while deep learning [23] used a 16 GB NIVIDA Telsa V100 hosted by AWS C2 GPU. It was reported that rule-based was implemented on an Arduino Due (Arduino, Somerville, MA, USA), a 32-bit microcontroller, Atmel SAM3X8E ARM Cortex-M3 processor with 512 kB flash memory, 96 kB SRAM and 84 Mhz clock speed. Multi-feature [22] scored lower than rule-based [21], which reported a relatively small latency.

Evaluating all the possibilities on based on the design matrix consideration factors, the most suitable AI algorithm for binary decision making was chosen. Single feature PR relationship between historical data and immediate measurement would be the best algorithm. Specifically, degree-of-polynomial (DoP) matching between PJ and IJ datasets was designed to quantitatively corelate immediate wellbeing of the subject. DoP was used as metric in quantifying time-series based PPG dataset. During operation of AHBT, the subject’s IJ dataset’s DoP would be compared to its PJ dataset’s DoP under the proposed ML algorithm. The algorithm would be able to determine with certain level of confidence if AHBT could proceed with patient transportation.

These operational perspectives could be depicted as shown in Figure 1. The proposed algorithm made use of PJ-DoP as reference to compare with the most recent IJ-DoP on condition of equality. There would be multiple matching attempts due to progressive IJ datasets generated over time. If both DoPs were to match, patient was deemed to have a strong correlation with PJ state, suggesting that his wellbeing was normal. These stepwise matches would culminate towards the final arrival at destination. Under present concept where both PJ and IJ DoPs did not match, patient was deemed to have no correlation to PJ state. Under such uncertainty, the patient was to be transferred to the nearest pre-designated stop point for further intervention.

### 2.2. Experiment Proceedings and Data Collection

An experimental run was designed to collect PPG data of individual subjects in a simulated environment similar to patient during hospital bed transport (HBT). This was based on Singapore hospitals’ settings which were often multi-building and multi-storey institutions where delays factors included lift-use priority protocols and foot traffic peaking in tandem with daily activities. A nominal HBT could easily take up to 30 min. Therefore, duration of each run was set at 1 h which covered a 5-min PJ phase, followed by 45 min time frame under simulated HBTs to study the efficacy of ML algorithm with enough data buffer. Posture of the subject was interchangeable between sitting or lying down in bed which were representative of HBTs scenario. Total freedom of the subject’s upper limb and wrist movements were allowed to capture naturally occurring movements. Also, subjects could be either awoke, resting or sleeping.

Experiment proceedings and data collection were approved by Nanyang Technological University’s Institutional Review Board (NTU-IRB), reference number IRB-2019-09-012. A total of 35 subjects (1 subject per run) were studied under central limit theorem guidance (*N* ≥ 30) to meet statistical significance. Healthy subjects aged 21 years and above were recruited for this voluntary study, as shown in Table 3.

During data collection, two Empatica E4 wristbands were worn simultaneously on the subject’s left and right wrists as shown in Figure 2. The E4 wristbands consisted of three physiological and one motion sensor to provide inferential physiological information about its wearer, summarized in Table 4. First was a PPG sensor measuring the wearer’s blood volume pulse (BVP) from the wearer’s wrist to indicate cardiac activity used in this study. Second was electrodermal activity (EDA) which measured wearer’s skin conductivity indicative of sweat levels as an implied indicator to wearer’s stress levels. Third, an infrared thermopile measured wearer’s skin temperature to give an indication of bodily heating and cooling. Last was a three-axis accelerometer which measured bodily movement of the wearer’s wrist as an indication of physical activity. Wristbands were turned on to initiate the start of PJ phase and were allowed to run for the entire duration of one hour. At end of each experiment, wristbands were switched off and data were uploaded using E4 Connect via E4 manger software. At all times, subject was accompanied by an immediate NTU research team member in a laboratory setting.

The ML algorithm was applied during post-analysis using Python’s sci-kit-learn. Standard excel plotter and matplotlib Python software were used for data and result visualization. Signal segmentation [26] was used to segment E4’s single and continuous PPG signal into distinct waveforms for assembling into PJ and IJ datasets. A DoP search range of 1st to 20th DoP was chosen based on a pilot study [20] to obtain the best possible quantifications using least amount of computing resources.

## 3. Results and Discussion

This section presents observations and analysis on algorithm’s goodness-of-fit, central tendencies, efficacy. From a total of 35 subjects, there were 70 PJ and 11,940 IJ datasets post-segmentation and each dataset was only used within its originating run. These findings covered both left and right wrists unless specified otherwise. There was no cross referencing between subjects.

### 3.1. Goodness of Fit

Goodness of fit was defined as quantification of how well a statistical model associated to its ground truth or dataset. Ground truths were information from direct observation. For PJ and IJ datasets, a perfect goodness of fit signifies a high-level matching between the patient PPG measurements and the modelled regression function. Hence, decisions can be made through the modelled function with references to the ground truths.

Likewise, a poor goodness of fit can be obtained through this decision-making process. For our proposed PR ML algorithm, the goodness of fit metric, cross validation score mean (CVSM), was calculated based on the average modeling of each dataset for 5 times between 1st and 20th DoP to aggregate potential ML permutation effects. All results from modelling across 1st to 20th DoP were analyzed.

The top five and worst five CVSMs across from left and right wrists are presented in Figure 3 for PJ datasets and Figure 4 for IJ datasets. Green crosses represent individual datapoints and the red line represents a regressed PR ML model.

Top 5 PJ datasets were consolidated between 0.2 and 0.6 CVSM at 8th to 10th DoP, under the T subplots in Figure 3. This was due to PR ML treating every datapoint with equal significance. This resulted in the regression ML model being represented within the major PJ datapoints, as shown in the red line of L-1T subplot in Figure 3. This suggested that in general, PJ’s definition of 5 min measurement time was adequate to trace the general PPG waveform trend.

The worst five PJ datasets were consolidated between near-zero to small magnitude negative CVSM at 18th to 20th DoP, as shown in the W subplots under Figure 3. This was due to mandatory PR ML modelling at sub-optimal DoPs. In particular, (L-1T, L-2W), (R-2T, R-1W) and (R-3T, R-3W) PJ subplot pairs in Figure 3 showed best and worst CVSM instances coming from the same subject’s PJ dataset. The proposed ML algorithm had demonstrated that it would only pick out the best goodness of fit outcomes.

An exception in PJ instances was from Subject 32 with 0.903 CVSM at 9th DoP, as shown in L-1T subplot in Figure 3. CVSM was high into 0.90 due to a highly harmonized body of PPG waveforms. This was a positive outlier representing a near-idealized outcome.

IJ datasets used continuous measurement which was different from PJ’s 5 min measurement window. The difference in data definitions would then be analyzed and compared with PJ outcomes.

The top five IJ datasets were consolidated between 0.967 and 0.994 CVSM at 9th to 10th DoP, as shown in all T subplots in Figure 4. This was due to IJ datasets containing less data by definition, which increased the chance of datasets with tightly grouped PPG waveforms. In general, this shorter time frame dataset definition was able to trace its PPG waveform trend. The resulting IJ DoP range also coincided with PJ’s DoP range, which was good indication for both DoPs’ matching.

In contrast to the earlier PJ datasets, the worst five IJ CVSM were consolidated near-zero or negative CVSM hovered between 1st and 4th DoP, as shown in all W subplots in Figure 4. This was also due to distinct abnormalities present in the IJ dataset. This resulted in more occurrence of 1st DoP where the regressed function was a linear line and even at 2nd and 4th DoPs when regressed function had included curvatures.

Comparing both PJ and IJ duration differences, a shorter time would result in instances of exceptionally good CVSM due to ease in assembling a coherent sets of PPG waveforms. However, the probability of low CVSM datasets also increased resulting in a wide CVSM spread when compared to PJ. This meant that shorter measurement duration comes with a cost of general difficulty to quantitatively converge a patient’s condition into a single mode. Patient wellbeing monitoring would then face increased difficulty by design to address multiple modes correlation to a single patient safe condition.

Conversely, a longer time would then result in a narrower CVSM window due to a larger probability in amassing a coherent body of PPG waveforms that can outweigh stray waveforms. However, mean CVSM was relatively lower due to minor variations within the coherent body of PPG waveforms. This meant that AHBT decision would be more spaced apart to achieve a more stable wellbeing monitoring. However, waiting too long to arrive at a decision would risk patient uncertainties during AHBT journey.

Moreover, other modes of CVSM extremities were observed and they will be discussed with further analysis in the next section.

### 3.2. Central Tendency

In this study central tendency served as an analysis on whether algorithm was stable or not. Ideally, only one mode should be associated with the highest goodness of fit to reduce uncertainties in algorithm decision making. Visually, this should be seen with a single peak DoP distribution in contrast to multiple peaks across DoP domain, as shown in Figure 5. For PJ datasets, single peak DoP would mean convergence of the subject’s known priori across a statistically significant group. For IJ datasets, a single peak DoP convergence coinciding with PJ’s DoP peak would mean high algorithm efficacy. Moreover, CVSM spread per DoP from Figure 6 to Figure 7 served as indication on level of data accountability where the ideal was for tight CVSM grouping with large number of occurrence and high mean at DoP distribution peaks. Every occurrence of peak DoP value would be a good validation of high algorithm stability across the population.

PJ and IJ DoP distributions of highest CVSM per dataset were shown to have coinciding single peak DoP distributions, shown as discrete histograms in Figure 5. It was observed that PJ distribution peaked between 7th and 8th DoP and IJ distribution also peaked at 8th DoP. In earlier pilot study using finger clip PPG sensor, peaks showed a single dominant DoP as it was easily achievable on a single subject [20], when compared to wristbands, two DoP peaks in PJ datasets were observed due to the change in sensor form factor and wider study population. Nonetheless, coinciding peaks meant PJ and IJ DoPs were desirable to be used for correlation and that inherent algorithm design and definitions were sound to represent patient condition.

Extending DoP search to 50th DoP showed no other mode of DoP peaks existed as there were no significant occurrence beyond 20th DoP. For PJ, lowest DoP occurrence was from left wrist single instance at 4th DoP while highest was also single instance but from right wrist at 16th DoP. Whereas for IJ, lowest DoP occurrence was 133 instances at 1st DoP while the highest was a single instance from right wrist at 40th DoP. The original search boundary of 1st to 20th DoP was adequate to model PPG data for both PJ and IJ datasets.

Single peak DoP distribution study was applied uniformly across 35 subjects with peaks observed between 7th and 10th DoP without any peculiarities in PJ datasets as shown in Figure 8. The algorithm could be used on a wider demographic, regardless of age.

CVSM spreads for PJ and IJ dataset’s DoP were shown in Figure 6 and Figure 7, respectively. In PJ dataset, both left wrist and right wrist’s CVSM spreads for 7th and 8th DoP had 11 occurrences, but with different values. The 9th DoP was not considered as it only had 2 instances. Even though 7th DoP was more consistent by its tighter CVSM spread, 8th DoP had a higher mean which indicates better data accountability, as previously discussed in Section 3.1. Compared to 7th DoP, the 8th DoP would be considered as dominant PJ DoP because it had a higher mean with a good number of occurrences. For IJ datasets, 8th DoP was considered to be the dominant DoP due to highest number of occurrences.

CVSM spreads were greater in IJ datasets as compared to PJ datasets, as shown in Figure 7. This was due to a lighter dataset definition of 30 PPG waveforms coupled with natural signal artefacts resulting in CVSM extremity potential, as discussed earlier in Section 3.1. This was much less in comparison to PJ dataset definitions, which on average comprised of 400 waveforms based on nominal 80 bpm heart rate within a 5 min measurement window. Nonetheless, higher CVSMs, especially 0.50 and above, were possible with at peak DoPs, between 8th and 11th DoP.

### 3.3. Efficacy: Degree-of-Polynomial (DoP) Match Rate

Efficacy was defined as the ratio of positive matches to match attempts between PJ and IJ datasets within a subject. An efficacy of 0 would mean the algorithm did not allow the hospital bed to proceed with the journey at all, while 1 would mean the algorithm would allow AHBT to successfully complete its journey.

For each match attempted, the binary output was either 0 or 1 to indicate either a negative or positive match. As shown earlier in Figure 1, a 0 output would bring the patient to a pre-designated stop point while a 1 would confirm that the patient could continue with AHBT journey. As discussed in Section 2.1, each subject used their own PJ dataset for its own PJ DoP generation. All subsequent IJ datasets and DoPs were then analyzed within the same subject’s PJ dataset and DoPs for efficacy calculations. During a match attempt, PJ and IJ DoPs must be the same for a positive match as equality based DoP matching was used. This would indicate a strong correlation between patient immediate wellbeing with a recent known safe state.

Our algorithm performed well to pick up all PPG signals which resulted in mean efficacy of 0.20 in Figure 9, which meant on average, AHBT completed 20% of its journey to destination. Furthermore, efficacies across both wrists showed similar distribution which suggested a potentially economical implementation that one sensor can be used on any wrist during an AHBT. This was typical for a conscious subject with added freedom of movement provided PPG signal variabilities over the one-hour experimental runs. In actual scenario of AHBT application, the PPG signal would be more stable as the patient was in either in unconscious or sedated state. Thus, this efficacy outcome was acceptable as conscious subjects’ dataset were employed in an AHBT application proof of concept development.

There was no peculiar trend observed that would suggest age as a factor on efficacy in Figure 10. Therefore, an eventual AHBT application could opt for deployment of a single wristband on either wrist for a wider age group. This observation was useful as alternative measurement points had become possible and simultaneous use of both wrists did not give any added advantage.

## 4. Conclusions

A first of its kind patient remote automated monitoring (RAM) PR for automated hospital bed transport (AHBT) was proposed and conducted using machine learning (ML) algorithm in this paper. The concept of autonomous patient monitoring was not new, but to apply it for AHBT would require a higher level of understanding and appreciation of the on the topic of Dreyfusian descriptor of a *Novice* nurse. Dreyfusian concept was used in designing three essential constituents of the algorithm. These included historical data where algorithm characterized patient’s known status 5 min just before the start of an autonomous transport journey. Second, immediate measurements during autonomous transport journey were taken and recorded. Finally, a binary decision condition based on ML algorithm was employed to control AHTB’s advance or stoppages.

Various AI algorithms with potential for AHBT application were considered using a nurse centric design matrix evaluation system. PPG was chosen as the parameter for wellbeing monitoring due to its proven deployment in hospital settings and privacy protection. Both wrists were measured simultaneously to study need for multiple sensor measurements. Data collection was done with a statistically significant population (*N* ≥ 30) who were above the age of 21. Studies showed that both wrists exhibited similar results.

Results were analyzed through goodness of fit, central tendency and efficacy analysis. High goodness of fit was observed in 0.2 to 0.6 cross validation score mean (CVSM) occurring at 8th to 10th DoP for PJ datasets and 0.967 to 0.994 CVSM at 9th to 10th DoP for IJ datasets, which meant the proposed algorithm was able to model PPG waveform trends. With the worst goodness of fit PJ consolidated at (0 CVSM, 20th DoP) and IJ (0 CVSM, 1st DoP) demonstrated CVSM to be a reliable metric to pick out best PJ and IJ DoPs for further analysis and AHBT decision making.

Central tendencies of both PJ and IJ DoPs coincided between 7th and 10th DoP. For PJ datasets, 8th DoP was considered to be the dominant DoP due to higher number of occurrences compared to 9th DoP and higher mean compared to 7th DoP. For IJ datasets, 8th DoP was considered to be the dominant DoP due to highest number of occurrences. CVSM spread were different between PJ and IJ DoPs, where PJ ones showed tighter convergence but lower CVSM mean in comparison to IJ datasets. This was due to dataset definition differences in PJ’s 5 min data sets as compared to the shorter IJ’s 30 PPG waveforms. Nonetheless, the coinciding DoP distribution showed a good stability qualities of our proposed algorithm.

Mean efficacy was 0.20 due to algorithm’s performance in capturing PPG signal variations from a conscious subject with freedom of movement. This was acceptable as a first AHBT proof of concept as most HBT were meant for unconscious or sedated patients where PPG signals were expected to be more stable.

The presented findings are important as a first ML proof of concept for AHBT application. Improving algorithm development is set for future work. These include using the same data definitions for PJ and IJ datasets, but with further implementation of a DoP tolerance band during DoP matching to increase efficacy; or developing acceptable standards to ensure CVSM values consistently lies in a specific window before algorithm decisions can be put to use. This study has further demonstrated the potential of time savings of nurse productive manhours up to 45 min for each HBT. These savings can be used towards more holistic patient caring tasks that would accelerate career progression of nurses towards advanced practice nurses (APN).

Extending this concept further towards in-community healthcare, a nurse can now better assess authenticity of patients’ near real time health data remotely to facilitate remote healthcare delivery. The use of commercial off-the-shelf sensors demonstrate technology readiness with more work to be done on algorithm in realizing a full RAM solution.

## Figures and Tables

**Figure 1 sensors-21-05711-f001:**
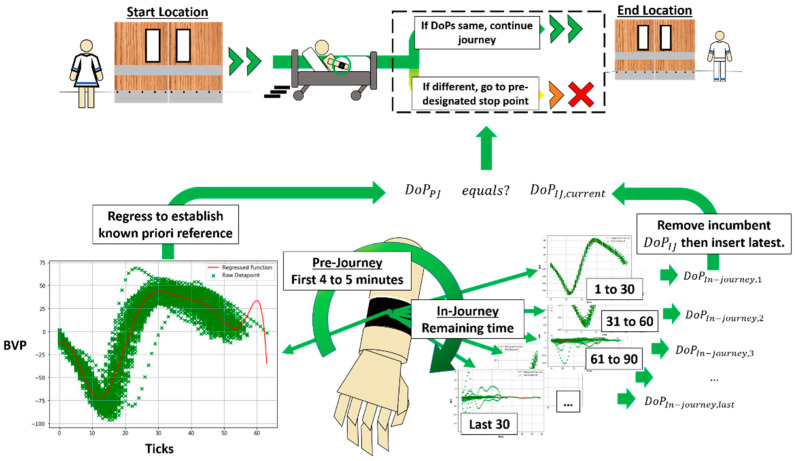
Schematic of photoplethysmography (PPG) remote automated monitoring (RAM) machine learning (ML) algorithm proposed for autonomous hospital bed transport.

**Figure 2 sensors-21-05711-f002:**
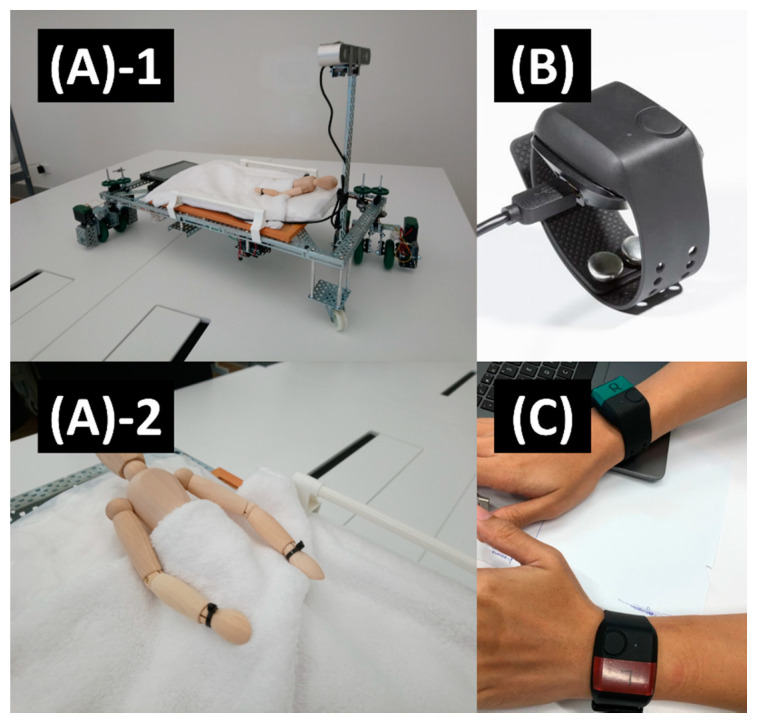
Scale down model of an autonomous hospital bed transport prototype was designed and tested. It included a set of motorized caster wheels and various autonomous navigation elements (**A-1**). Prototype mockup of a patient wearing PPG wristbands on both wrists simultaneously (**A-2**). Example of both PPG wristbands worn simultaneously on both hands of a subject using Empatica E4s (Empatica Inc., Boston, MA, USA) (**B,C**).

**Figure 3 sensors-21-05711-f003:**
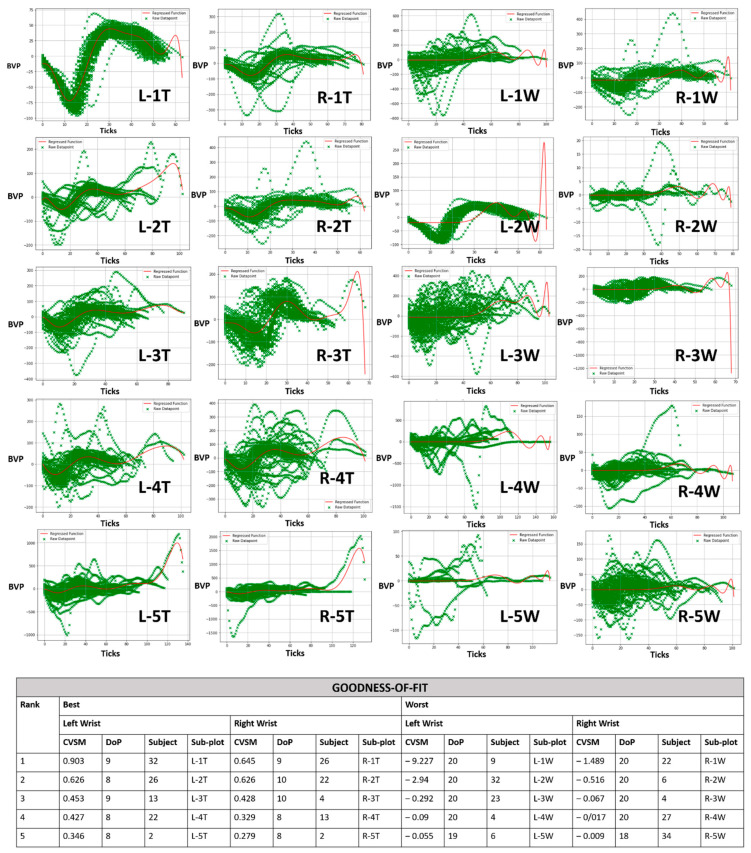
Goodness-of-fit of pre-journey (PJ) datasets ranked by top five and worst five CVSMs. Each sub-plot consists of an Empatica E4 wristband’s blood volume pressure sensor value (BVP) versus ticks (sensor time interval). [Left wrist = L and right wrist = R].

**Figure 4 sensors-21-05711-f004:**
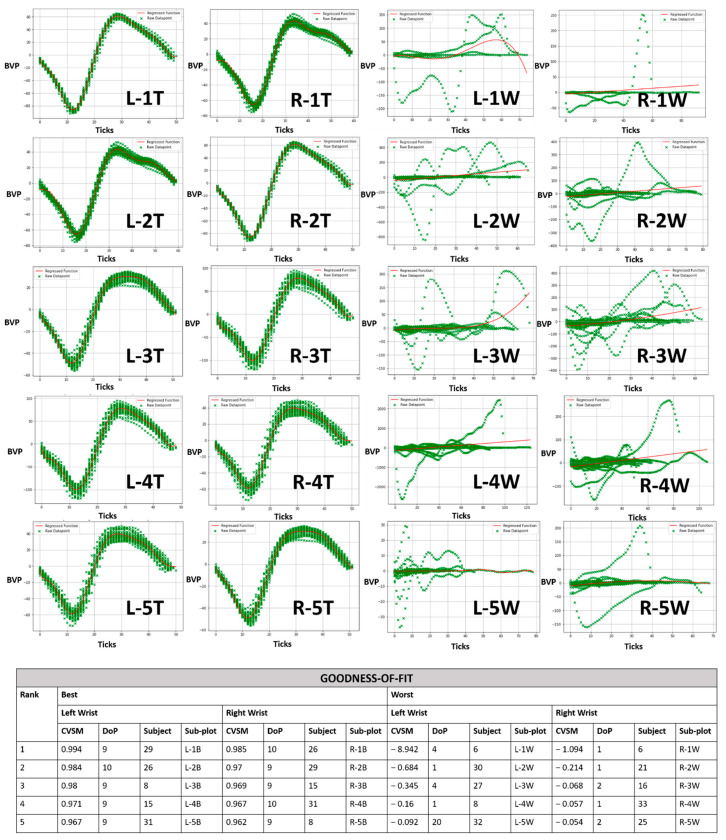
Goodness-of-fit of in-journey (IJ) datasets ranked by top five and worst five CVSMs. Each sub-plot consists of an Empatica E4 wristband’s blood volume pressure sensor value (BVP) versus ticks (sensor time interval). [Left wrist = L and right wrist = R].

**Figure 5 sensors-21-05711-f005:**
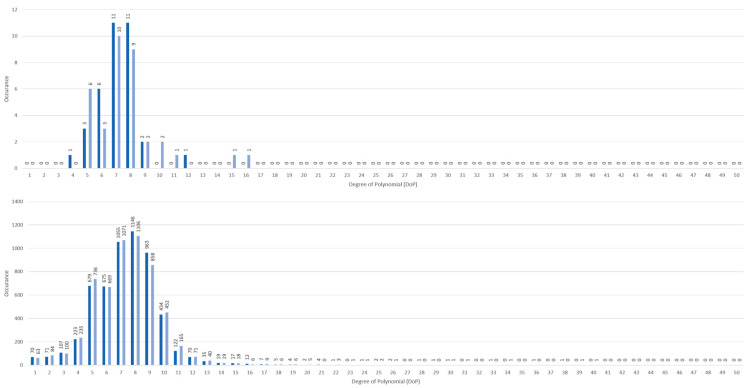
DoP distribution for PJ datasets (**top**) and IJ datasets (**bottom**). Only highest CVSM per dataset was registered.

**Figure 6 sensors-21-05711-f006:**
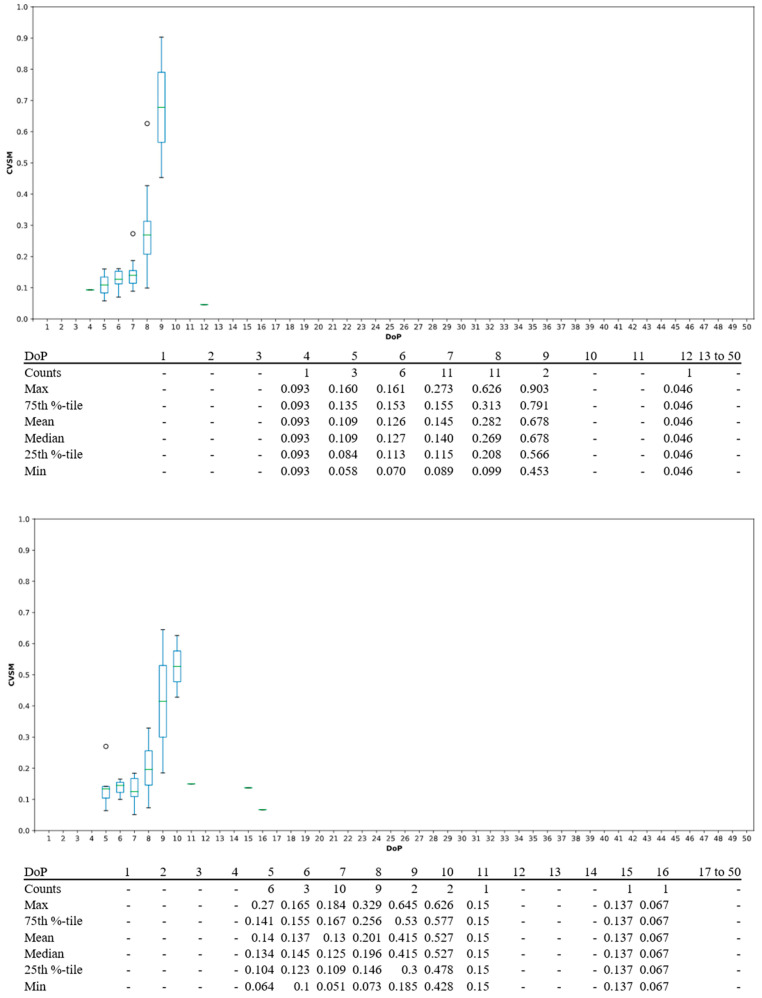
CVSM spread per DoP for PJ datasets. Left wrist (**top**) and right wrist (**bottom**).

**Figure 7 sensors-21-05711-f007:**
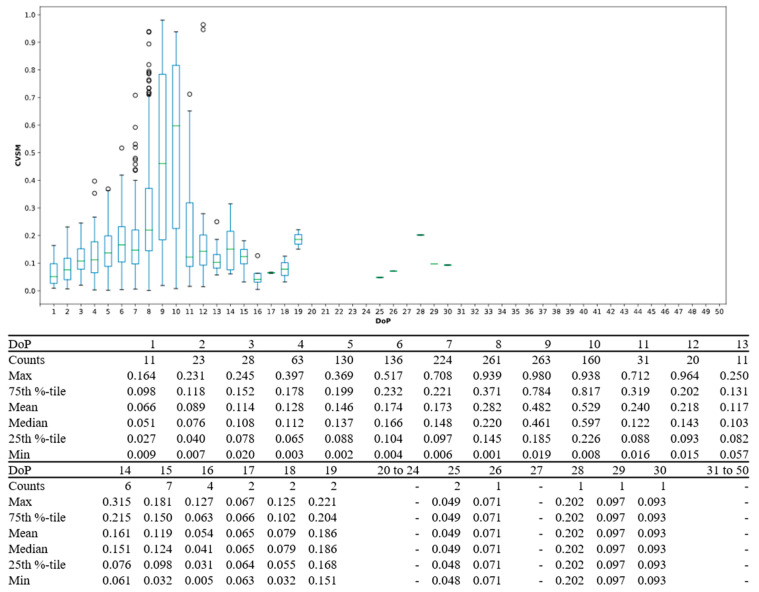
CVSM spread per DoP in IJ dataset. Left wrist (**top**) and right wrist (**bottom**).

**Figure 8 sensors-21-05711-f008:**
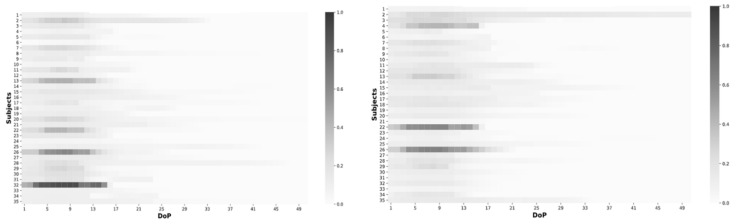
Heatmap distribution of PJ DoP by each subject for all 35 subjects. Left wrist (**left**) and right wrist (**right**).

**Figure 9 sensors-21-05711-f009:**
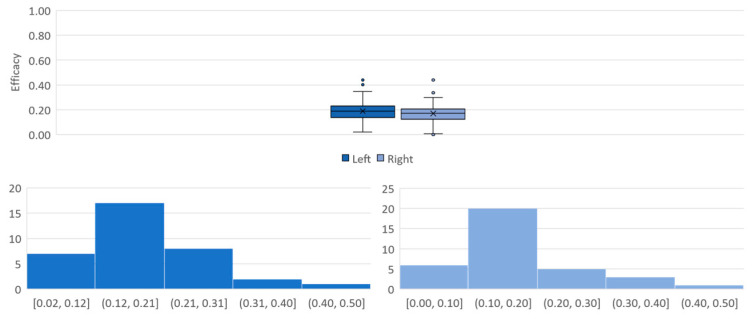
Efficacy spreads for each wrist (**top**). Efficacy distributions left wrist (**bottom-left**) and right wrist (**bottom-right**).

**Figure 10 sensors-21-05711-f010:**
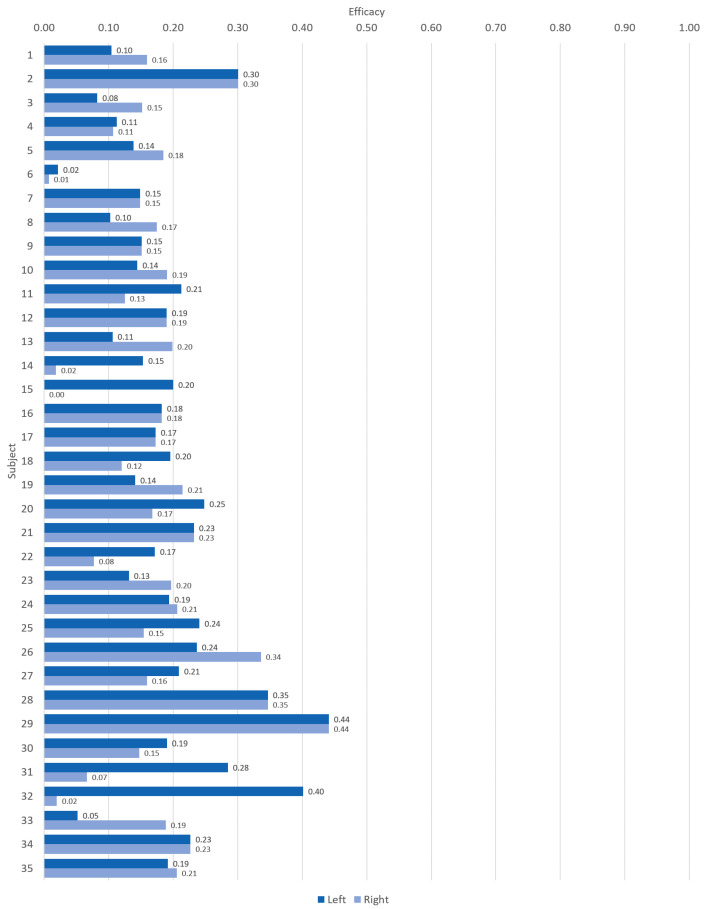
Efficacy for each subject.

**Table 1 sensors-21-05711-t001:** Translation of Dreyfusian descriptor to proposed photoplethysmography (PPG) remote automated monitoring (RAM) machine learning (ML) algorithm constituents.

Algorithm Constituent	Historical Data	Immediate Measurement(s)	Binary Decision Making Condition
Dreyfusian Descriptor (Analogy) [13]	“To determine fluid balance, check the patient’s morning weights and daily intake and output for the past three days.”	“Weight gain and an intake that is consistently higher than output…”	“By greater than 500 cc. could indicate water retention, in which case fluid restriction should be started until the cause of the imbalance can be determined.”
This work: Autonomous Hospital Bed Transport (AHBT)	Pre-journey dataset (PJ)	In-journey dataset (IJ)	PJ-IJ feature matching to determine if AHBT can continue journey or move to a predesignated stop location.
Dataset or Technical Specification(s)	Supervised and segmented PPG waveforms from 5 min just prior to AHBT. PPG data authorized by supervising nurse.	Remotely supervised and segmented sets of PPG waveforms. Sets are latest and non-overlapping of 30 PPG waveforms.	Performed by the most suitable PR ML technique chosen by design matrix evaluation.

**Table 2 sensors-21-05711-t002:** Design matrix to select suitable AI algorithm technique for AHBT.

Factor	Weight	Single-Feature PR [20]	Rule-Based [21]	Multi-Feature [22]	Deep Learning [23]
Easy to understand	30%	25	15	10	10
Explicability	20%	15	10	10	5
Not using data sets beyond the individual subject	20%	20	20	20	0
No data annotation	20%	15	20	15	10
Computation time	10%	7	9	8	5
Total Score	100%	81	75	64	30

**Table 3 sensors-21-05711-t003:** Dataset of human subjects breakdown by their ages.

21 up to <30	30 up to <65	65 or Older
27	7 *	1 ^#^

* (Subject number, age): (18, 30), (2, 33), (24, 48), (8, 49), (29, 51), (26, 57), (23, 60). ^#^ (Subject number, age): (31, 71).

**Table 4 sensors-21-05711-t004:** Empatica E4 wristband sensor breakdown.

Sensor Included in E4	Physiological Phenomena	Information Inferred
Photoplethysmography	Blood volume changes	Cardiac activity
Electrodermal activity	Skin’s electrical conductivity	Sweat (Stress) levels
Infrared thermopile	Skin’s thermal conductivity	Body heating and cooling
Three axis accelerometer	Wrist bodily movements	Wearer’s physical activity

## Data Availability

Not applicable.

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
