# Peer review of "Application of a Machine Learning Algorithms in a Wrist-Wearable Sensor for Patient Health Monitoring during Autonomous Hospital Bed Transport"

_sensors, 2021, doi:10.3390/s21175711_

Round 1

Reviewer 1 Report

This work developed a polynomial regression (PR) machine learning (ML) algorithm for remote automated monitoring, i.e., Hospital bed transportation, in order to relieve the nurse resource from in-person patient monitoring. The author compared different ML algorithm in terms of the difficulties to be understood by nurse with basic mathematic training, easy to implement in miniaturized computer, etc., and chose PR as the suitable one to fit these desired features.

few questions are:

  • What is the physics behind the PPG and the automation of hospital bed transportation?
  • Why the experiments are designed to be one hour? As a more practical point of view, would a bed transportation be finished in few minutes? Or is the transportation between diff hospitals?

Author Response

The authors deeply appreciate the feedback and comments from Reviewer 1.  We've included the summary of the reviewers comments and response in the following word file.  Thank you very much once again.

Reviewer 2 Report

Authors propose a polynomial regression machine learning remote automated monitoring (RAM) algorithm for wellbeing monitoring for the autonomous hospital bed transport (AHBT) application. This proposed method has advantages such as simplicity and quick computation. Tests were used into both wrists of 35 subjects. This manuscript requires several modifications based on the following comments:

1.-Introduction section. This section must include the main advantages and limitations of the proposed method in comparison with other reported in the literature.

2.-Authors should include more discussion about the results indicated in Figures 3, 4, 5,6, 7, 8 9 and 10.

3.-References must be modified to the format of Sensors.

4.-Authors should include more detail information about the main parameters of the sensors and characteristics of the 21 patients. In addition, this manuscript can include more information of the noise in the measurements.  

Author Response

The authors deeply appreciate the feedback and comments from Reviewer 2.  We've included the summary of the reviewers comments and response in the following word file.  Thank you very much once again.

Reviewer 3 Report

The manuscript reports the application of photoplethysmography (PPG) remote automated monitoring (RAM) algorithms based on Dreyfusian descriptor for the monitoring of automated hospital bed transport (AHBT). The algorithm quantified Pre-journey and In-journey PPG datasets to decide whether to continue AHBT by checking the degree of polynomial between quantified datasets. Although the concept of AHBT is not new, the Dreyfusian descriptor used for the RAM algorithm seems interesting, and also the manuscript is written well. Overall, the work can be considered for publication in this journal after addressing the following concerns.

Comments:

  1. To measure the goodness of fit, is there any reason to measure the pre-journey data for a specific measurement window for matching PPG and regression function? What will be the impact on wellbeing monitoring if PPG data were taken for a shorter/longer time or in different volunteers postures?
  2. The visible and bigger size of the text along abscissa and ordinate in Figures 1, 3, and 4 would be better for the clarity of readership.
  3. How the cross-validation score mean (CVSM) can be improved for efficacy for better decision if there is no harmonization in data measurement? For example, larger variation in negative values of CVSM for W-subplot can lead to efficacy in favor of binary 0 to stop the AHBT journey.

Author Response

The authors deeply appreciate the feedback and comments from Reviewer 3.  We've included the summary of the reviewers comments and response in the following word file.  Thank you very much once again.
